# Graphene Biosensors—A Molecular Approach

**DOI:** 10.3390/nano12101624

**Published:** 2022-05-10

**Authors:** Mónica Machado, Alexandra M. L. Oliveira, Gabriela A. Silva, Diogo B. Bitoque, Joana Tavares Ferreira, Luís Abegão Pinto, Quirina Ferreira

**Affiliations:** 1Instituto de Telecomunicações, Avenida Rovisco Pais, 1049-001 Lisbon, Portugal; alexandra.oliveira@nms.unl.pt; 2iNOVA4Health, CEDOC Chronic Diseases Research Center, NOVA Medical School, Universidade Nova de Lisboa, Campo Mártires da Pátria 130, 1169-056 Lisbon, Portugal; gabriela.silva@nms.unl.pt (G.A.S.); diogo.bitoque@nms.unl.pt (D.B.B.); 3Faculdade de Ciências Médicas, Nova Medical School, Universidade Nova de Lisboa, Campo Mártires da Pátria 130, 1169-056 Lisbon, Portugal; 4Ophthalmology Department, Centro Hospitalar Universitário de Lisboa Norte, 1649-035 Lisbon, Portugal; joana.t.ferreira@chln.min-saude.pt (J.T.F.); l.pinto@campus.ul.pt (L.A.P.); 5Visual Sciences Study Centre, Faculty of Medicine, Universidade de Lisbon, 1649-028 Lisbon, Portugal

**Keywords:** graphene, biomolecules, self-assembly, graphene-oxide, nanomedicine, biodevices

## Abstract

Graphene is the material elected to study molecules and monolayers at the molecular scale due to its chemical stability and electrical properties. The invention of scanning tunneling microscopy has deepened our knowledge on molecular systems through imaging at an atomic resolution, and new possibilities have been investigated at this scale. Interest on studies on biomolecules has been demonstrated due to the possibility of mimicking biological systems, providing several applications in nanomedicine: drug delivery systems, biosensors, nanostructured scaffolds, and biodevices. A breakthrough came with the synthesis of molecular systems by stepwise methods with control at the atomic/molecular level. This article presents a review on self-assembled monolayers of biomolecules on top of graphite with applications in biodevices. Special attention is given to porphyrin systems adsorbed on top of graphite that are able to anchor other biomolecules.

## 1. Introduction

Graphene is considered one of the most important materials of the century, having numerous applications in areas including electronics, medicine, agriculture, structural engineering, and many others that are under investigation. Graphene is attractive due to its flexibility, elasticity, hardness, resistance, and biocompatibility, being inert on one hand but easily functionalized, if required, and having high thermal and electrical conductivity properties on the other. In addition, it is transparent and nonreactive in the presence of ionizing radiation. Recently, graphene has been explored has a potential material for biomedical applications in areas such as biosensing, cancer therapies, drug delivery systems (DDS), and tissue engineering, allowing the development of new research fields associated with the interactions of biomolecular systems and graphene. The interactions between biomolecules and the flat surface of graphene have mainly been studied at the molecular level, mostly of them by using scanning tunneling microscopy (STM) to evaluate the molecular conformation and organization as well as the conductivity of single molecules, the molecular mechanical properties, and the stability of specific molecular systems [1]. 

Two types of molecular interactions with graphene systems have been addressed: physically adsorbed systems on graphene and molecules covalently linked to graphene oxide (GO). The oxide form of graphene, known as GO, is usually used to attach molecules on its oxygen functional groups, allowing the study of covalently linked molecular systems. GO is an interesting material due to its biocompatibility and ease of use in surface functionalization. For these reasons, GO is the material elected to study biomolecules that can be attached directly to the epoxy or hydroxyl groups existing on its surface or attached to other molecules working as anchor points to the GO surface [2]. 

Physically adsorbed systems, such as self-assembled monolayers, are a promising method to build two-dimensional organized surfaces with important applications in organic electronics. STM is the technique of choice to study systems with atomic and molecular resolution. At the liquid interface it is possible to “observe” the molecular organization on top of graphene in real time and to induce molecular organization through the solvent, tip voltage, and temperature. Usually, physisorbed systems on graphene are governed by π-π interactions and are established by aromatic rings that contain π orbitals. These interactions are very promising for several applications, because they are spontaneous, contribute for the self-organization of molecular systems, and are nondestructive and reversible. The development of materials for bioapplications has been greatly advanced due to these types of interactions, mainly for biosensor applications, nucleic acid sequencing, crystal packing, protein folding, and DDS, among other multilayered film systems containing graphene [1]. 

This article reviews the most recent studies on biomolecules on top of graphene surfaces and the potential applications of these biomolecules with a focus on self-assembly systems. The main applications of graphene biosensors are summarized and include electrochemical, field-effect transistor, mid-infrared plasmonic, optical, and fluorescent biosensors. Finally, a future perspective of the potential of these materials is given. 

## 2. Self-Assembled Biomolecular Systems on Top of Graphene

Graphene and its derivatives have attracted increasing research interest due to their ordered honeycomb lattice pore structure, high surface area with a great number of active sites, mechanical strength, high thermal conductivity, and great electrical and optical properties. GO, with its two-dimensional network and heterogeneous chemical and electronic structure, has gained particular interest [2,3,4]. However, sometimes, it is necessary to functionalize graphene to enhance its properties and to improve the adsorption of biomolecules on its surface. That functionalization is achieved with organic groups like hydroxyl, carboxyl, or amino. Graphene derivatives, such as GO and reduced graphene oxide (rGO), are structurally composed with functional groups that improve its interaction with other molecules. The two-dimensional structure containing oxygen on the surface and sp^2^ domains allows for interaction/anchoring with a wide range of biomolecules, such as DNA, peptides, proteins, including enzymes, viruses and others, via covalent and noncovalent interactions, as seen in Figure 1. The functionalization of graphene with these biomolecules has promising applications in the areas of materials science, nanotechnology, and biomedicine [3,4,5,6].

Many properties of graphene nanomaterials are due to the oxygen functional groups, hydroxyl, and epoxy spread over the surface of GO; therefore, these constitute the most important part of the functionalization of graphene and its interactions with other molecules.

The anchoring of biomolecules allows the active sites to be accessible to target molecules as well as preserve their conformational stability and bioactivity. For good immobilization, it is important to look at the properties of the biomolecule, like its functional groups, size, and polarity [4]. 

Graphene/biomolecule systems have numerous applications in relevant fields, like biology and materials science. They can be used in DDS with a vehicle function to load and transport drugs into target cells and release these drugs through a specific process, and also as a biosensor that can be either an electrochemical, field-effect transistor (FET), mid-infrared plasmonic, optical, or fluorescent biosensor [6]. 

### 2.1. DNA

DNA is a crucial molecule that stores and transfers genetic information by acting as a template for RNA synthesis. Due to its biocompatibility, renewability, and helix structure, DNA is an important biomolecule for the functionalization of graphene materials. The graphene–DNA link can be achieved through noncovalent interactions (π-π interactions, electrostatic interactions, and hydrogen bonding). As demonstrated by Zhang and co-workers, DNA can be immobilized on the graphene surface by electrostatic interactions between the functionalized positively charged graphene and the negatively charged nucleotides [6,7]. Noncovalent interactions are unstable in the presence of surfactants and some complicated solvents. Therefore, many studies have been conducted to immobilize DNA on the surface of graphene by covalent interactions, as shown by Zhang et al. [4,6,8]. The authors used a DNA probe strand immobilization platform from a stable and well dispersed graphene–pyrenebutyric acid (PBA) nanocomposite to anchor DNA [6,8]. In another approach, Wang et al. tried to conjugate DNA and GO nanosheets through a copper-catalyzed azide–alkyne reaction. In their study, they achieved a stable system with an extraordinarily high DNA density [6,9]. Bo et al. designed an electrochemical DNA biosensor through the immobilization of DNA on a bioelectrode (glassy carbon electrode modified with layers of GO and polyaniline nanowires) [4,10].

### 2.2. Proteins

Proteins are large-sized, complex molecules composed of one or more amino-acid chains. They are present in all living organisms and have various functions, such as catalyzing metabolic reactions, structural functions, regulating tissues and organs, replicating DNA, responding to stimuli, and transporting molecules between locations. 

Similar to DNA, proteins can be attached to graphene by covalent and noncovalent interactions. To anchor proteins on graphene covalently, the GO nanosheets have to be functionalized first. However, Hui et al. stated that proteins can easily be adsorbed on GO nanosheets using different interactions with hydrophilic adsorption favored by the hydrophobic aromatic groups on the GO surface [6,11]. Due to the structure of amino acids in proteins, they can be anchored to the graphene surface by π-π stacking. Noncovalent interactions are made through intermolecular forces, dipole-dipole interactions, and opposing surface charges. Aromatic residues of proteins interact with graphene’s aromatic structure by hydrophobic interactions. Myung et al. reported a biomolecular sensor based on graphene-coated nanoparticles (NPs) for the selective and sensitive detection of proteins as biomarkers for breast cancer. First, they functionalized silicon oxide NPs with 3-aminopropyltriethoxysilane, and then these were coated with thin layers of GO, preventing particle aggregation while maintaining high electrical conductivity. The nanoparticles were then adsorbed onto GO layers through electrostatic interactions between the negatively charged groups of GO and the positively charged groups of silicon oxide NPs [12]. 

### 2.3. Peptides

Peptides are short strings of amino acids called the “building blocks” of proteins. They are smaller than proteins, and the main differences between them are their sizes and structures, with peptides having less-defined structures. Natural peptide monomers can be directly attached to graphene through covalent interactions. For example, Kanchanapally et al. constructed 3D GO-based membranes with an antimicrobial peptide [6,13]. The bonding between peptides and graphene/GO can also be governed by noncovalent interactions, particularly through π-π stacking, due to the affinity of the different amino acids and their locations in the peptide sequence [6,13]. Wu et al. synthesized a GO-peptide hybrid hydrogel, linking peptide monomers directly to graphene without using a crosslinker. Modified peptides containing a GO binding motif (pyrene) were linked to GO sheets by noncovalent interactions (π-π stacking) [6,14]. These noncovalent interactions are favorable for the design of biosensors with diagnostic functions [4]. 

Wang and Lin proposed a nonenzymatic, electrochemical hydrogen peroxide sensor, using a glassy carbon electrode modified by GO-peptide-Silver Nanoparticles (AgNPs) [4,15]. Recently, self-assembled peptide nanofibers (PNFs) sparked scientific interest because of their versatile applications as building blocks. For example, Li and co-workers developed a nonenzymatic electrochemical H_2_O_2_ biosensor based on silver nanowires (AgNW) to form the nanocomposite rGO-PNF-AgNW. The development was made in four steps, namely through the synthesis of rGO-PDDA(poly(diallydimethylammonium chloride), the creation of PNFs, the deposition of AgNPs on the PNF surface, and the binding of PNF-AgNW (silver nanowires) nanohybrids onto the rGO surface [6,16].

### 2.4. Viruses

Viruses are composed of nucleic acids (DNA or RNA) covered in a thin coat of protein; thus, the graphene–virus interaction essentially occurs through noncovalent interactions, such as hydrogen bonding, electrostatic interactions, or redox reactions [6]. Graphene–virus bonding can be useful in many applications. For example, Lee et al. designed a highly selective, ultrathin membrane though GO nanosheets and unidirectional, aligned M13 viruses using negatively charged GO nanosheets. The linkage between GO and M13 viruses was ruled by strong electrostatic forces allied with auxiliary hydrogen bonding among the carboxylate-functionalized groups of GO and the virus ends. The neutral parts of M13 viruses were attached to the GO sites that were chemically less active, and they were immediately aligned by an external force applied by directional washing. These aligned viral layers provided a highly size selective, thin nanomesh membrane with great permeation properties [6,17]. Another example of a GO–virus complex application was given by Oh et al., in which M13 viruses were used on a conducting framework. The authors’ approach had the purpose of enhancing the interactions between viruses and graphene through a structural modification of the major coat protein of the M13 virus. This modification allowed the virus–graphene linkage to occur through π-π interactions, and with the Hopp–Woods scale, it was demonstrated that viruses and graphene can interact by hydrophobic–hydrophobic interactions [6,18].

### 2.5. Enzymes

Enzymes are biomolecules (typically proteins) that act as highly selective catalysts of the chemical reactions of a biological system, accelerating the rate and specificity of metabolic reactions ranging from the digestion of food to muscle and nerve function and DNA synthesis. They are composed of long chains of amino acids, organized in a 3D structure. Because of the enrichment of oxygen groups, GO is the perfect substrate to immobilize enzymes on its surface, as it does not require any modifications or functionalization. Interactions between GO and enzymes can be achieved through covalent forces, noncovalent π-π stacking, and electrostatic interactions. Covalent bonding is the most commonly used method for the immobilization of enzymes on the GO surface. Unnikrishnan et al. described an easy way to immobilize glucose oxidase (GOx) on rGO, in which free amino groups of GOx link covalently with edge-plane functionalized groups of GO [19]. Noncovalent interactions, specifically electrostatic forces, are another way of immobilizing enzymes on top of graphene. Li et al. used this principle to fabricate a GO-enzyme nanohybrid for bacterial detection and drug screening. They proved that GO-ethylenediamine (EDA) interacts with β-galactosidase (β-Gal-an anionic enzyme) by electrostatic forces [6,20]. 

Due to the large surface of graphene and its excellent properties for immobilizing enzymes, many studies on the use of GO–enzyme complexes have been conducted, for example, in the medical field as biosensors or in the environmental field with applications on water treatments. One example of a GO–enzyme complex in biosensors was given by Liu et al., who presented a glucose biosensor based on an oxidoreductase enzyme (GOx) which can catalyze the oxidation of glucose to hydrogen peroxide (H_2_O_2_) [21]. In the first step, rGO was functionalized with chitosan (Chi) and was then self-assembled with GOx by electrostatic forces in aqueous solution, obtaining rGO@Chi-GOx nanostructures that act as a biosensor with a good response to glucose [6,21]. An example of a GO immobilized enzyme used in water treatments was demonstrated by Zhang et al., in which horseradish peroxidase (HRP) was immobilized on the GO surface through electrostatic interactions for application in the degradation of various phenolic compounds. The obtained HRP-GO system showed great thermal stability and a wide active pH range. This system was tested in seven phenolic compounds, and the results showed a strong removal efficiency compared with the use of HRP alone [6,22].

### 2.6. Porphyrins

Porphyrins are highly colored organic compounds composed of four pyrrole-type subunits connected by methinic bridges in a cyclic model with great applications in the medical field, especially in cancer therapy, due to their unique physical, chemical, and spectroscopic properties. Because of their noninvasive nature, porphyrins have generated great interest as an efficient alternative to chemotherapy [2].

Porphyrins are being used to functionalize GO to obtain nanohybrid platforms with special photoactive properties. Porphyrins can interact with the GO surface through electrostatic interactions; however, covalent interactions may occur between the –CO_2_H groups of GO and the –NH_2_ groups of porphyrins, like the condensation reaction of 5,10,15,20-tetrakis(4-aminophenyl)porphyrin (TAPP) with the GO-activated group –CO_2_H. Another approach was made by Santos et al., in which GO was linked directly to porphyrins previously functionalized with different glycol branches. In the first step, the authors performed carboxylation of GO with chloroacetic acid under basic conditions to maximize the –CO_2_ groups, and these were subsequently linked (by ester bonds) to the functionalized porphyrin groups through covalent interactions [2,23]. This GO/porphyrin system was successfully tested in tumor cells, and its biocompatibility with the human Saos-2 osteosarcoma cell line was established, showing that it is a great approach for cancer therapies [2].

As mentioned before, cationic porphyrins and GO suspensions can bond through electrostatic interactions; however, these interactions are pH-dependent. For example, Larowska et al. demonstrated that GO sheets have weaker linkages to 5,10,15,20-tetrakis(4-trimethylammoniumphenyl) porphyrin (TMAP) at an acid pH when compared with interactions at a pH of around 7 [24]. In another study, Shim et al. tested the role of electrostatic interactions between GO and 5,10,15,20-tetrakis (1-methyl-pyridinium-4-yl) porphyrin (TMPyP) using an acid-base approach and electronic spectroscopy. From his experiments, Shim concluded that electrostatic interactions are extremely important to initiate and preserve the GO/Porphyrin system, while π-π stacking is mainly present in the molecular flattening of the TMPyP molecule [2,25]. 

Porphyrins are also able to organize themselves into different supramolecular structures when conjugated with GO [23]. An example of this property was demonstrated by Ma et al., who prepared a thin film based on TMPyP and surface-modified GO through electrostatic and π-π interactions achieved by the Langmuir–Blodgett technique. The authors proved that TMPyP molecules were adsorbed on floating layers and self-assembled into highly ordered J-aggregates [26]. Another application of a GO/porphyrins system is in the early diagnosis of cancer by the identification and quantification of cancer biomarkers. Stefan van Staden et al. studied this approach using exfoliated graphene (E-NGr) modified with protoporphyrin as a sensor for the recognition of gastric cancer biomarkers (carcinoembryonic antigen—CEA, carbohydrate antigen 19–19—CA 19–19, and protein tumor suppressor—p53) in the blood and urine of patients [27].

### 2.7. Compilation of Graphene/Biomolecule Systems 

A summary of self-assembled biomolecular systems on graphene is given in Table 1. It shows the types of interactions between the biomolecules and graphene. These can be divided into covalent and noncovalent interactions. Noncovalent interactions are mainly π-π interactions, electrostatic forces, and hydrogen bonds, and although noncovalent bonds can occur spontaneously, these interactions are weaker than covalent interactions.

Therefore, to design a better system, it is crucial to know the structure and properties of each biomolecule to select the most appropriate anchoring approach [6]. 

## 3. Self-Assembled Biomolecular Systems on Top of Graphite

Self-assembled structures on graphite, such as monolayers or supramolecular structures, are of particular interest due to their auto-organization ability, allowing highly reproducible fabrication. Despite their self-organization, the formation of a self-assembled system on graphite is dependent on several factors, such as the functionalization of the substrate, the solvents used, the temperature, the number of components that adsorbed on the surface, and the existence of an external stimulus that can interfere with the adsorption. The functionalization of graphite to anchor biomolecules is a challenge considering that no reactive groups are present. Recently, with the increasing interest in the use of graphene in DDS or in biosensors to identify drugs, the anchorage of nanocarriers on the top of graphite has been exploited. Cyclodextrins (CDs) are an example of biomolecules that have attracted particular attention from researchers. CDs are a family of cyclic oligomers with a hydrophobic cavity and a hydrophilic shell. The nomenclature and cavity size of CDs depend on the number of glucose units in the structure. All three forms, namely α-CD (6 units), β-CD (7 units), and γ-CD (8 units), have similar structures. They are homogeneous, crystalline, and nonhygroscopic in nature and have a hydrophobic inner cavity and a hydrophilic exterior. However, β-CD has a higher number of hydroxyl groups and a lower solubility due to the internal hydrogen bond network that is created between the secondary hydroxyl groups. β-CD enhances the water solubility and bioavailability of several substances, allowing for the incorporation of drugs into aqueous vehicles and therefore being of great pharmacological interest.

The geometric shape and hydrophobic cavity of CDs make them ideal to accommodate various hydrophobic guest molecules in aqueous environments, giving them applications in molecular recognition, drug delivery, explosive detection, water purification, and chiral separation. CDs can also be applied in sensors, allowing for the formation of complexes with electroactive guests; however, this requires special conditions to immobilize the host molecule.

De Feyter et al. demonstrated a simple and fast one-step covalent immobilization of β-CD on a highly oriented pyrolitic graphite (HOPG) surface. On their study, the authors used diazonium salt from aniline-modified β-CD that was electrochemically grafted on HOPG by the formation of C–C bonds between aryl radicals and sp^2^ carbon on the surface. The complex β-CD-HOPG was characterized by STM, atomic force microscopy (AFM), and Raman spectroscopy. The electrochemical behavior of the developed modified surface was tested by cyclic voltammetry towards ferrocene, hexycyanoferrate, and the neurotransmitter dopamine (DA), and it was used as a sensor for the detection of these substances. The sensor exhibited excellent sensitivity for DA detection at the nanomolar level, higher than the results reported at the time for a 3D nitrogen-doped graphene/β-CD electrode and comparable to the highest sensitivity reported for any electrochemical dopamine sensor [28].

The STM is one of the techniques elected to analyze biomolecules on top of graphite, since it allows for not only the observation of molecules adsorbing onto a surface but also the promotion or manipulation of the adsorption. This ability enables the design of complex and customized structures starting from single atoms or molecules [29]. For this reason, STM has been used to characterize and build controlled self-assembly structures. Ferreira et al. described a supramolecular system composed of 23 units of porphyrins and bipyridines linked one-by-one using STM in real time and at room temperature. Molecular wires, 14 nm in length, were prepared in a step-by-step manner with 13 layers of zinc(II)-octaethylporphyrin (ZnOEP) alternating with 12 layers of 4,4′-bipyridine on top the HOPG surface (Figure 2). The self-assembly of each layer was monitored in real time by STM imaging, and current measurements of the porphyrin/bipyridine wires were performed. This imaging control is essential to confirm the molecular organization of each monolayer before proceeding to the assembly of the next one [30].

STM also enables the achievement of supramolecular systems through bottom-up approaches at the solid/liquid interface based on atom-by-atom or molecule-by-molecule assembly working inside of a drop. The molecules present in a solvent drop are adsorbed to form one or more monolayers, whose growth is monitored by STM high-resolution imaging. Different types of molecules can be adsorbed using well-defined specific conditions, such as the nature of the substrate and solvent, the operating temperature, the concentration, and the characteristics of the tip (Figure 3). This method can be important in the design and fabrication of biosensors using graphite as an electrode considering that multiple molecular systems can be added until the desired structure is achieved.

STM at the solid/liquid interface allows for the visualization of the dynamics of the self-assembled system formation by controlling the formation of a monolayer to achieve a highly stable molecular organization. To study the different formation stages of monolayers on a substrate, Fichou et al. used STM in real time and evaluated the structural evolution of Hexakis(n-dodecyl)-peri-hexabenzocoronene (HBC-C_12_) on monolayers of n-pentacontane (n-C_50_H_102_) on the HOPG surface. The STM images showed the formation of three distinct phases during HBC-C_12_ packing at the molecular level. Self-assembled monolayers of n-C_50_H_102_ were deposited onto the HOPG surface from a solution of n-tetradecane. To the HOPG- n-C_50_H_102_ modified surface, HBC-C_12_ was added as a saturated solution of n-tetradecane. STM images were recorded for eight hours, revealing three successive distinct phases: the α-phase (oblique structure), β-phase (dimer structure), and γ-phase (row structure). The α-phase formed in the first hour, and the transition to a more stable β-phase occurred during the second hour after the deposition of the HBC-C_12_ molecules and was completed in two/three hours. After five/six hours of HBC-C_12_ deposition, the final transition from β to γ-phase occurred and was completed in two more hours. With this experiment allied with STM images, it was possible to analyze and characterize the monolayer’s phases during the formation steps. It was demonstrated to be a possible way to control the structure of a molecular 2D lattice [32].

The formation of monolayers on a substrate depends on the surface diffusivity and the adsorption rate, and the ratio between them defines the molecular packing. Ferreira et al. used STM at a solid/tetradecane interface to analyze the dynamics involved in the zinc(II)-octaethylporphyrin (ZnOEP) monolayer formation on the HOPG surface in real time. Self-assembled monolayers were formed by the addition of a porphyrin-saturated solution of n-tetradecane to a n-tetradecane droplet (on HOPG surface) [33]. The rearrangement of the molecules and the monolayer formation were monitored by STM imaging, and the results showed the clear formation of a polymorphic monolayer that was completed after different stages. When ZnOEP was added to a n-tetradecane droplet, a metastable α-phase was formed, and in the following two hours, thermodynamically stable β-phase nucleation centers started to appear. After this nucleation time, the full conversion from the α to the β-phase was completed in one hour. The polymorphic behavior that was observed demonstrated that the adsorption of ZnOEP on the HOPG surface is ruled preferentially by diffusion mechanisms over the adsorption rate [33].

At a liquid/solid interface, STM allows for the manipulation of reactions using the tip voltage as an initiator. De Feyter et al. studied this principle by applying a voltage pulse on the STM tip to a manganese chloride-porphyrin (Mn(III)-PP) monolayer on a HOPG substrate. The authors observed the formation of bright spots during the application of voltage pulses to the STM tip at different places. With the topographical images, apparent heights, and the oxygen dependence observed, the authors concluded that these bright spots represent µ-oxo dimers, a characteristic oxidation product of the reaction of Mn-porphyrin with oxygen. Furthermore, it was proved that it is possible to induce a local reaction on a surface by applying a voltage pulse on the tip of the STM [34].

Another demonstration of STM’s versatility is the possibility of analyzing the individual electrical properties of molecular systems using scanning tunneling spectroscopy (STS). Matsuda et al. measured molecular wires attached to 2D phase-separated templates for the evaluation of single molecular conductance. STM allowed the evaluation of the single molecular conductance of two molecular wires with distinctive alkyl side chain lengths on the HOPG surface [35]. For this study, two different tetraarylporphyrin-Rh(II) complexes with different side chain lengths were synthetized, and C_22_-Rh-1 and C_30_-Rh-2. STM measurements were performed at the 1-octanoic acid/HOPG interface with a constant current mode. The concentrations of the two porphyrins were optimized for a stronger affinity to the HOPG surface with a 20 times higher concentration for C_22_-Rh-1. After this optimization, the tip was immersed into the sample solution on the HOPG surface, and images were collected. With the obtained images, the apparent height of the two porphyrins was determined by section analysis and histograms that were fitted with a single Gaussian function. The results of seven scans showed a difference in the apparent height between the two porphyrins of around −1.10 ± 0.55 Å. Since the aim was to compare the single molecular conductance of molecular wires with different lengths, Equation (1), which expresses the conductance ratio (G1/G2) as an exponential function of the apparent heights (Δh_stm_) and molecular heights (Δx) was applied.
(1)$$\frac{G1}{G2}=\exp\;\{\alpha ({\Delta h}_{\mathrm{stm}}-\Delta x)\}$$

Molecular heights (Δx) were obtained by considering both the s-cis and s-trans conformations of the two wires using the density function theory (DFT) calculation. The conductance ratio of C_22_-Rh-1 and C_30_-Rh-2 was estimated to be 1.3 ± 0.7, a value that is in accordance with previously reported data obtained by the mechanically controllable break junction (MCBJ) method. With STM studies, the authors concluded that the apparent height ratio and conductance ratio showed a straightforward correlation for a single layer of molecules [35].

## 4. Graphene and Bioapplications in Biosensors

In the last decade, graphene and its derivatives have generated great interest in the development of biosensors for different applications, from the diagnosis and treatment of diseases to the detection of enviromental contaminants or applications in the food industry, as schematized in Figure 4.

A graphene-based biosensor is composed of three main parts: the analyte, which is the target molecule to be detected; the bioreceptor, which is a biomolecular system that is able to receive/detect/identify the target molecule; and a transducer, which is responsible for converting the biorecognition occurrence to a signal that can be measured and which is proportional to the concentration of the analyte. Several sensor types can be adopted: Electrochemical, Mid-Infrared Plasmonic biosensors, Fluorescent graphene-based biosensors, and Optical and field effect transistors (FET) biosensors.

Electrochemical biosensors are the most commonly used type based on GO due mainly to the hydrophilicity of GO, which enables different biomolecules and nanomaterials, such as quantum dots (QDs), semiconducting NPs, polymers, and metals to adsorb onto its surface. This enhances electron transfer between the bioreceptor and transducer, which is advantageous for electrochemical sensors with high sensitivity [4]. Graphene also plays an important role in Mid-Infrared Plasmonic biosensors. These sensors are based on the principle of light absorption by the biomolecules, which is detected through their vibrational fingerprints and their refractive index. Graphene and its derivatives have the ability to capture light, and when interacting with biomolecules, they exhibit their characteristic properties. In Mid-Infrared Plasmonic biosensors, the Surface plasmon resonance (SPR) technique is used to study biomolecular interactions in real time without supplemental steps, and it is common to use gold films as conduction bands. Singh et al. presented a highly efficient SPR immunosensor for the biotinylated cholera toxin. To amplify the SPR signal, nitrilotriacetic acid was attached to graphene through π-π interactions with pyrene derivatives, which were secured by eletropolymerization. They successfully obtained an immunosensor for the specific antibody anticholera toxin [4,36]. Fluorescent graphene-based biosensors were shown to successfully diagnose various types of cancer and cardiovascular disease using their biomarkers, and ultrasensitive detection of biomolecules in the early phases of several diseases and even health monitoring could also be achieved. These biosensors are based on the characteristic fluorescence intensity of some molecules that contain fluorophores, such as proteins, NPs, or small molecules. With the appropriated wavelength of light, it is possible to excite fluorophores, causing a change in the fluorescence intensity that is detected by the sensor, generating a measurable value [4,37]. The principle of biosensors relies on the transformation of the biological reaction between the target analyte and sensing elements (e.g., biomolecules) into detectable signals. These signals can be related to the human body, for example, through pulse, heart rate, body motion, the blood oxygen level, blood pressure, glucose, cholesterol, or cancer cells, as well as to toxins in food products and heavy metals in drinking water. Bioreceptors, such as antibodies, enzymes, or nucleic acids, are usually immobilized on graphene surfaces in order to allow the targeting molecules for interactions [6,38]. Optical and FET graphene-based biosensors are being studied due to their high sensitivity in detection. Optical biosensors detect changes in the absorption of UV/visible/Infrared light that are related to the interaction of microorganisms/biomolecules with the analyte. This is based on the measurement of photons involved in the process. Graphene has unique optical properties, broadband and tunable absorption, and strong polarization-dependent effects, which are ideal for graphene-based optical sensors [4]. FET biosensors have great advantages considering that they do not need florescent labels or electrochemical tags [4,39]. They are solid-state sensors where the electroconductivity of the semiconductor between the source and drain terminals is regulated by a third gate electrode through an insulator. To promote the detection ability, biomolecules are attached to the sensing channels. Changes in the signal interactions between these attached biomolecules and target species are detected and converted into measurable signals. All of these sensors depend on the ability of the bioreceptor to detect the analyte.

The challenge is to attach the bioreceptor to the graphene surface (transducer) to ensure the correct analysis of the target molecule. The most common bioreceptors used on graphene transducers are DNA, antibodies, and enzymes. The method used to immobilize these systems is known, and it was reported that DNA and antibodies are fixed on the top of graphene via 1-ethyl-3-(3-dimethylaminopropyl)carbodiimide (EDC)/N-hydroxysuccinimide (NHS) chemistry and the enzymes are physisorbed. The following sections highlight the three main types of bioreceptores: enzymes, nuclei acids, and antibodies.

### 4.1. Enzymes as Bioreceptors

Enzymes are commonly fixed on top of graphene to detect glucose, phenols, cholesterol, glutathione, and hydrogen peroxide (H_2_O_2_) [38,40]. For enzyme detection, Nicotinamide adenine dinucleotide (NAD) is used as a standard. This coenzyme can have two oxidation states, namely NAD^+^ and NADH, which are present in several enzymatic reactions. NADH, one of the most important coenzymes in the cells, is involved in metabolic processes, and this is the reason why NAD is used as the standard. Cheraghi et al. demonstrated the efficacy of glutathione peroxidase in biological fluids. Glutathione peroxidase is an enzyme that is involved in glutathione metabolism, being the ideal marker for glutathione detection. In the work of Cheraghi et al., glutathione peroxidase was adsorbed to the GO surface covalently via EDC and NHS onto a glassy carbon electrode (GCE) functionalized with GO and nafion polymer. The resulting biosensor demonstrated great sensitivity to glutathione and was successfully used in real samples with acceptable data and results confirmed with statistical tests (F-test and *t*-test) [41].

Glucose biosensors are still under development to increase their sensitivity and effectiveness. The blood glucose level is the most important parameter for the diagnosis and management of diabetes, one of the most common diseases worldwide. Therefore, it is crucial to monitor this pathology to avoid severe comorbidities. For the detection of glucose levels, the enzyme glucose oxidase (GOx) is used, and significant effort is being made to enhance the electrical contact of redox enzymes with the surface electrodes of enzyme-based electrochemical glucose sensors. Qi et al. immobilized gold nanoparticles (AuNPS) on GO surfaces via a benzene (Ph) bridge with aryldiazonium salt. The hybrid system GO-Ph-AuNPS was then incorporated into a 4-aminophenyl modified GCE. After that, GCE/GO-Ph-AuNPS were functionalized with 4-carboxyphenyl (CP), and finally, GOx was attached through covalent interactions to obtain the glucose sensor GCE/GO-Ph-AuNPS-CP/GOx. The sensor demonstrated a good range of sensitivity for the detection of glucose levels as well as good selectivity results [38,42]. Glucose biosensors are based on the oxidation of glucose to GOx with the glucose levels measured by the differential between the drain-source current and the Dirac point shift of the graphene transistor. You et al. applied this principle to create a silk fibroin-encapsulated graphene FET enzymatic biosensor for glucose detection, and silk protein was used in both the device’s substrate and enzyme immobilization material. It showed a large linear detection range and excellent selectivity and sensitivity [6,43]. Wei et al. improved the electrochemical performance by developing a glucose sensor with high sensitivity [44]. In this case, Graphene layers were directly grown on a substrate by chemical vapor deposition (CVD), improving the electrochemical performance with the elimination of impurities. Construction of this sensor was achieved by the direct deposition of graphene layers on a silicon oxide/silicon (SiO_2_/Si) substrate without a transfer process, eliminating residue contamination by polymer transfer-support and heavy metal ions. After this process, GOx was sequentially immobilized, and the polymer nafion layer was deposited on the functionalized graphene surface (by oxygen-plasma treatment). Another example was reported by Liu et al. where GOx enzymes were immobilized on carboxyl acid (COOH)GO groups through covalent interactions and amide bonds in the presence of EDC and NHS. This biosensor showed good sensitivity and exhibited biocompatibility with human retinal pigment epithelium cells [38,45].

Cholesterol sensors are also based on the enzymatic detection approach. Cholesterol is a fat-like substance that is present in all cells and is an important building block of healthy cells. However, high levels of cholesterol can lead to the development of deposits in blood vessels, increasing the risk of heart disease. Therefore, the determination of cholesterol levels is important to help with the prevention of the development of cardiovascular diseases, such as atherosclerosis, hypertension, cardiopathy, and myocardial infarction.

Cholesterol biosensors must be able to detect the precise level of cholesterol; therefore, many studies have been focused on achieving the desired sensitivity using the enzyme cholesterol oxidase (ChOx). For example, Dey et al. used an enzymatic detection approach in which they attached ferrocene redox units to the surface of GO through covalent interactions. The resulting cholesterol biosensor showed a good linear response, as well as good sensitivity [38,46].

In another experiment, Agnihotri et al. reported a nonenzymatic cholesterol sensor based on β-CD functionalized graphene. The authors proposed that β-CD could be linked to GO sheets covalently, forming the complex GO-β-CD, which has a good sensitivity range [38,47].

Enzymatic biosensors have been applied to detect H_2_O_2_, which is a subproduct of numerous enzymatic reactions as well as being an important product in the food industry that could have pharmaceutical and environmental applications, especially due to its oxidizing and reducing properties. For the detection of glucose, cholesterol, and H_2_O_2_ levels in blood, Li et al. developed tyramine (TYR)-functionalized Graphene Quantum Dots (TYR-GQDs), in which tyramine is adsorbed on the GQD surface by covalent interactions between the TYR amino groups (NH2) and the GQD carboxyl groups (COOH). The TYR-GQDs fluorescent sensor showed high sensitivity and selectivity [4,48]. These types of biosensors were improved by Settu et al., who presented an electrochemical sensor based on laser-induced graphene electrode and multiwalled carbon nanotubes (MWCNT) for H_2_O_2_ detection. The incorporation of MWCNT improved the performance of the sensor due to the enlarged electroactive surface area and decreased charge-transfer resistance at the laser-induced graphene-electrolyte interface. This enhanced sensor showed better sensitivity than those without MWCNT [49].

Graphene biosensors using enzymes as bioreceptors are also available to detect pesticides. Pesticides are substances used to control pests in agriculture and include herbicides and insecticides. Their excessive and uncontrolled use can cause food and environment contamination and, therefore, illness in humans and animals.

To prevent toxic effects, it is imperative to detect contaminants from pesticide residues in food and water. Hondred et al. presented an electrochemical sensor PtNP-IML-PGE/GA (Platinum Nanoparticles-Inkjet maskless lithography-patterned graphene electrode functionalized with glutaraldehyde) for the detection of organophosphorus pesticides (Ops), which are insecticides used to protect crops. The authors electrodeposited PtNP on patterned graphene electrodes (IML-PGE) to improve the electrical conductivity, electrocatalytic activity and surface area. After this step, the phosphotriesterase enzyme was conjugated with the functional cross-linking molecule glutaraldehyde (GA). The final biosensor showed a quick response to the insecticide paraoxon (a type of organophosphate) with high sensitivity and a low detection limit [38,50].

Graphene quantum dots (GQDs) have been widely used in the development of biosensors for the detection of pesticides applying the Förster resonance energy transfer (FRET) mechanism due to their unique fluorescence quenching properties. Gao et al. presented a N-(aminobutyl)-N-(ethylisoluminol) (ABEI) functionalized GQDs biosensor with great selectivity for the detection of multiple pesticides. This biosensor can detect different pesticides, such as flubendiamide, thiamethoxam, dimothoate, and chlorpyrifos [51].

A recent approach was made by Wang et al. to develop a highly sensitive and selective OP biosensor improved by the immobilization of acetylcholinesterase (AChE) on the gold gate electrode with chitosan (Chi), as schematized in Figure 5. AChE is an enzyme involved in biological nerve conduction that can hydrolyze acetylcholine into choline and acetic acid in order to manage acetylcholine levels in the body. Chi has been increasingly applied in biosensors due to its antibacterial properties and nontoxicity. In this case, the sensor was based on a graphene channel and source, drain, and gold gate electrodes. The amino groups of Chi form stable covalent bonds with the –COOH groups of AChE, improving the immobilization of AChE on the gold gate. The sensor was shown to have a good selectivity level and recovery performance in rice samples. However, the performance of the sensor was limited by the relatively low enzyme activity after immobilization and the poor conductivity of the gate electrode [52].

The unique properties of graphene-base nanomaterials make them ideal for the development of biosensors. In Figure 4, different types of biosensors that can be developed with graphene-base nanomaterials are presented.

### 4.2. Dopamine, Ascorbic Acid and Uric Acid Biosensors

Dopamine (DA) is a neurotransmitter present in the brain that acts as a chemical messenger between neurons. It is involved in motivation, memory, attention, and even in the regulation of body movements. Ascorbic acid (AA) is a natural water-soluble vitamin that is a potent reducing and antioxidant agent. It has many functions, such as detoxifying reactions, fighting bacterial infections, and forming collagen in fibrous tissue, teeth, bones, skin, and capillaries.

Uric acid (UA) is a natural waste product from the digestion of purine foods. A high uric acid level normally occurs when kidneys are not capable of eliminating it efficiently.

The three substances DA, AA, and UA have electrooxidation peaks almost at the same potential values; therefore, several efforts have been made to separate their signal potentials with greater sensitivity and selectivity. To overcome issues with amplifying the detection signal, graphene and GO can be doped with nitrogen, polymers, different inorganic metals, and metal oxide particles. An example of this approach was demonstrated by Srivastava et al., who proposed a functionalized multilayer graphene system to use on an amperometric urea sensor with high sensitivity [38,53].

To achieve a biosensor that is capable of detecting the vesicular secretion of catecholamine molecules (DA, epinephrine, and norepinephrine), He et al. developed a rGO FET with living neuroendocrine PC-12 cells. These cells are widely used in brain disease research.

rGO FET devices were functionalized in the first step with poly-L-Lysine, and then PC-12 cells were directly cultured on the top of rGO FET sensor. A solution with a high concentration of potassium ions(K^+^) was also used to depolarize the cell membrane and promote the influx of calcium ions (Ca^2+^) through the voltage-gated Ca^2+^ channels, as can been seen in Figure 6 [6,54].

Dopamine molecules and GO have a good affinity, which enables multiple noncovalent interactions, allied with fast decay at the picosecond range of the near-IR fluorescent. This affinity promotes the effective self-assembly of DA on GO’s surface. The fluorescent quenching between DA molecules and GO’s surface enables the development of a GO-based photoinduced charge transfer (PCT) label-free near-infrared fluorescent biosensor with direct readout of the near-IR fluorescence of GO for the selective and sensitive detection of DA. This sensor was presented by Chen et al. and was used for DA detection in biological fluids with quantitative recovery (98–115%) [4,55].

The neurotransmitter DA is also associated with Parkinson’s disease (PD), a neurodegenerative disease involving the progressive loss of nigrostriatal neurons. For this reason, the control of DA levels might be a biomarker for diagnosing the pathology in the early stages and also to optimize DA replacement therapy. To improve the sensor sensitivity, Butler et al. tuned the graphene surface chemistry with a simple one-step thermal annealing process with no need for subsequent chemical or biomolecular functionalization, as demonstrated in Figure 7. The authors used commercial graphene ink and colloidal suspensions of graphene flakes to improve the electrochemical response by controlling the mass transport. Graphene ink was deposited on a flexible polyamide substrate via spin-coating, and the annealing process was conducted at 300 °C for 30 min. The authors demonstrated that graphene ink can be used to develop low-cost and printable sensing devices with a high level of performance. The use of 2D inks allows for the formation of porous electrodes that offer an additional “catalytic” improvement over planar graphene sheets by modifying the mass transport properties of the electrode [56].

### 4.3. Nucleic Acids as Biosensors

DNA and RNA are nucleic acids that have distinct functions in living beings: DNA is responsible for storing and transferring genetic information, while RNA acts as a messenger between DNA and ribosomes to make proteins. Nucleic acid biosensors have attracted the attention of researchers due to their importance in gene engineering and in the diagnosis of genetic diseases. Many approaches, including fluorescence, electrochemistry, colorimetry, chemiluminescence, electrochemiluminescence, and SPR have been used to develop several biosensor types.

DNA and RNA have characteristic structures that are favorable to the establishment of π-π bonds between nucleobases, their conjugated systems, and graphene. However, the π-π interactions between nucleic acids and graphene are weak, and to improve them, Loan et al. deposited gold (Au) films on graphene layers, which modified them to enable stronger covalent interactions [38].

Huang et al. also used this Au advantage and presented a DNA biosensor (graphene/Au nanorods/polythionine) for the electrochemical detection of human papillomavirus (HPV) with a strong detection performance. The principle is based on the attachment of target DNA to the DNA structure of the capture probe on the electrode surface [38,57].

Recently, Bagherzadeh et al. studied the charge transfer and electrostatic potential of an armchair graphene nanoribbon (AGNR) to improve DNA hybridization detection. 1-Pyrenebutanoic acid succinimidyl ester (PBASE) binds to graphene surfaces through noncovalent forces via stacking of its pyrene group. Single-strand DNA (ssDNA) was immobilized on the AGNR surface by covalent interactions with PBASE. The bonding of probe DNA to PBASE was achieved with the formation of an amide bond, which resulted from the nucleophilic substitution of N-hydroxysuccinimide by an amine group of probe DNA. It was concluded that the functionalization of AGNR with PBASE increased the conductance of the sensor. The developed sensor showed a sensitivity of 10% at a zero bias voltage and a higher sensitivity with the application of a suitable gate voltage [58].

GO-based fluorescent biosensors have been widely used in research for the detection of biomolecules, such as DNA, RNA, and proteins. In many of these biosensors, DNA is used as a base matrix of the biosensor for target molecule recognition and reporting. An example of this usage was presented by Xing et al. through a fluorescent aptasensor based on a double-stranded DNA (dsDNA)/GO complex as the signal probe to study exonuclease I activities. dsDNA is adsorbed on the GO surface, allowing fluorescence quenching, and when the target molecule is introduced, the aptamer (short DNA folded molecule) preferentially binds with its target, which results in a nuclease reaction with a minor change in fluorescence when GO is introduced due to the weak affinity between generated mononucleotides and GO [4,59].

The detection of changes in living cells, namely transformation in cancer cells, has been a challenge; therefore, significant research effort has occurred, especially in the field of optical sensors. Wang et al. developed a graphene-based optical biosensor for the detection of drug responses in live cells with sensitivity for detecting cancer cell responses to anticancer drugs, as represented in Figure 8 [4,60].

Nucleic acid biosensors are used also to detect pathogens. Tiwari et al. developed an electrochemical biosensor iron oxide-Chi functionalized GO layers. The *E. coli* O157:H7 strain was immobilized on the GO functionalized surface through covalent interactions. The authors demonstrated that the incorporation of GO improved the electrochemical properties of the complex and the DNA detection of *E. coli*, resulting in a highly selective and sensitive biosensor [38,61].

Efforts have been made towards improving the sensitivity of biosensors to detect *E. coli* O157:H7. Jaiswal et al. used aminopropyltrimethoxysilane (APTMS)-functionalized zinc oxide nanorods (ZnONRs) and carboxylated graphene nanoflakes (c-GNF) that were electrophoretically deposited on an indium tin oxide (ITO) coated glass substrate and used as electrodes for the covalent immobilization of an *E. coli* O157:H7-specific DNA probe. The ZnONRs were functionalized with APTMS to incorporate amino groups on its surface, and then the electrostatic interactions with the carboxylic acid groups on c-GNF promoted the formation of an APTMS-functionalized ZnONR/c-GNF composite (APTMS-ZnONR/c-GNF). This composite was electrophoretically deposited on ITO substrate, and the final platform was immobilized with probe DNA sequences for the selective determination of *E. coli* O157:H7DNA using the impedimetric technique. The sensor retained its activity for five repetitive uses and with a low concentration detection [62].

Acid nucleic biosensors have also been used to detect toxic heavy metals, such as cadmium (Cd^2+^), chromium (Cr^3+^), lead (Pb^2+^), copper (Cu^2+^), and mercury (Hg^2+^), which are relatively dense metals or metalloids that are characterized by their potential toxicity. Heavy metal ions can enter and accumulate in the human body through food chains. Graphene and its derivatives have shown great efficiency in removing toxic heavy metals in aqueous solutions. Muralikrishna et al. had the objective of developing a sensor for the simultaneous detection of Cd^2+^, Cu^2+^, Pb^2+^, and Hg^2+^, and focused on the development of a L-cysteine/rGO based sensor. The GO surface was functionalized with L-cysteine through amidation between the GO carboxyl groups and L-cysteine amine groups and nucleophilic substitution in one step. The complex L-cysteine/rGO worked as a cathode with the ability to simultaneously detect all four heavy metal ions. This sensor showed detection limits for metal ions lower than the World Health Organization standards [38,63]. The development of a biosensor for toxic heavy metal ions is based on the interactions between metal ions and DNA base pairs (using DNA as a probe). For these interactions, graphene-based nanomaterials have gathered great interest as fluorescent probes to detect metal ions, such as Ag^+^, Pb^2+^, Hg^2+^, Cu^2+^, and F^3+^ (Fluor). Wen et al. developed a fluorescent nanoprobe based on GO and an Ag-specific nucleotide for the detection of Ag^+^ ions, and this showed good sensitivity [4,64].

### 4.4. Antibodies as Bioreceptors

Antibody bioreceptors have been implemented in cancer biosensors. Cancer is related to the uncontrolled division of cells that spread into surrounding tissues. It is considered one of the major causes of death worldwide. Detection and diagnosis of cancer in its initial stages is crucial for the success of treatments. Biomarkers are a helpful tool for early detection and even the prediction of cancer pathologies. Biomarkers are biological molecules found in the blood, body fluids, or tissues, which are suitable for presenting alterations of characteristic values or expressions. Specifically, cancer biomarkers are usually proteins, genes, and other molecules that affect the growth of cancer cells as well as how they multiply and how they die.

For all the reasons above, significant effort has been made to develop the ideal sensor for the detection of cancer biomarkers to detect or predict a person’s predisposition to develop a certain pathology.

Ren et al. doped GO with sulfur and attached polyaniline (PANI) to GO’s doped surface to prepare a biosensor for the detection of cancer biomarkers (carcino-embryonic antigen and nuclear matrix protein 22). The authors developed a tri-antibody dual-channel biosensing platform to increase the detection sensitivity of the two antigen types, as illustrated in Figure 9. The biosensor showed an improved sensing performance for both cancer biomarkers [38,65].

Recently, Zahra et al. presented a herceptin (HER2)-conjugated graphene biosensor that uses the special antibody of HER2, which turns this biosensor into an immunosensor for single-cell breast cancer detection. A graphene-based immunosensor was prepared on the surface of a GCE to quantify SK-BR-3 (human breast cancer cell line that overexpresses HER2 antibody) cancer cells. Selective tests showed excellent results in real samples and a linear range of 1–80 cells. The reproducibility of the biosensor was estimated to be about 66%. The stability and functionality range was about 40 days [66].

Graphene biosensors using antibodies have demonstrated successful results in food toxin analysis. Food toxins are natural substances generated by the metabolism of fungi, algae, plants, or bacteria that have negative effects on humans and animals. Examples of natural toxins in plant-based foods include those from the fungi Aspergillus flavus and Penicillium verrucosum. To prevent food contamination and subsequent illness, some studies have focused on the development of biosensors. Srivastava et al. developed an electrochemical biosensor for the detection of the food toxin AFB1. The authors used rGO sheets, and through covalent interactions, attached monoclonal antibodies to the toxin AFB1(anti-AFB1) to form the rGO/anti-AFB1 based sensor, which showed high sensitivity and stability for a period of 45 days. These improvements were due to the good electrochemical properties, large surface area, and fast electron transfer kinetics of rGO [38,67]. Aptamer-based fluorescence quenching sensing probes have triggered great interest for the detection of pathogens and food toxins. In experiments by Lu et al., a thiolated aptamer specific for AFB1 (aflotoxin B1, a secondary fungal metabolite of Aspergillus flavus) was adsorbed on the surface of QDs through ligand exchange. The differences in fluorescence between the aptamer-modified QDs and GO were registered. The authors presented a simple, sensitive, and selective sensor for the detection of AFB1, showing that this type of biosensor can be used to analyze other mycotoxins [4,68].

For food safety monitoring, Parate et al. wanted to develop a biosensor to detect histamine levels. Histamine (2-(1H-imidazol-4-yl)ethanamine) is a biogenetic amine present in seafood that is related to fish product spoilage and seafood allergies. To create the biosensor, the authors used the aerosol jet printing (AJP) technique, which is based on three main steps: graphene ink formulation, aerosol jet printing, and post-print baking. Graphene ink was aerosol jet printed into interdigitated electrode patterns and then baked at 350 °C in air circulation to improve graphene’s mechanical properties and to increase the electrical conductivity. In the final step, the devices were annealed in CO_2_ at 400 °C to functionalize the graphene surface with carboxyl and carbonyl groups, improving the interaction between the antibody and the graphene interdigitated electrode. The developed biosensor showed a wide histamine sensing range and a low detection limit [69].

Many studies have been carried to develop biosensors for the biomedical, clinical, and even environmental fields. To ensure enduring success, continuous improvement is needed, and therefore, new methods for graphene usage are still being studied. Table 2 summarizes the latest developments in GO biosensors.

## 5. Conclusions and Future Perspectives

As pointed out throughout this manuscript, significant developments have been made in the development of graphene-based nanomaterials by exploring their excellent properties that make them ideal for use in applications in a large variety of fields, from medicine to the environment. The good thermal and electrical conductivity of graphene along with its large surface area and numerous active groups makes it the ideal material for the immobilization of biomolecules for their application in biosensors.

This review has described the most recent advances in graphene-based nanomaterials as a platform for biomolecules. We presented the different interactions between graphene materials and biomolecules such as DNA, proteins, peptides, viruses, enzymes, and porphyrins as well as examples of their applications. A large number of graphene-based biosensors reported in the literature have shown good sensitivity in the targeting of biomolecules as well as good stability. To improve the sensitivity of biosensors, authors have functionalized the graphene surface with biomolecules or even nanoparticles, such as gold, obtaining excellent results. However, new functionalization methods for Graphene surfaces are needed to improve the immobilization of biomolecules. It is also important to improve the selectivity to detect and even predict severe pathologies in the initial stages. The early detection of some diseases is crucial to the success of treatments.

The detection of biomarkers is becoming the most important parameter to allow actions to be taken in a timely manner to avoid severe comorbidities and excessive spending costs on health and to provide relief for hospital resources and increase the quality of life of patients. One example was presented by De Feyter et al. with the development of a simple and fast single-step process for the immobilization of β-CD on the HOPG surface for the detection of DA, a neurotransmitter associated with Parkinson’s disease. Zahra et al. made another great advance by developing a biosensor for breast cancer cells that achieved good results. These two diseases are directly related to patients’ quality of life, and every step forward in their early diagnosis makes a difference.

Graphene and its derivatives also have great potential in applications involving drug delivery, and increasing interest in their applications in the biomedical field has emerged due to their exceptional physiochemical properties and unique planar structure. GO has a high drug-loading capacity, because of its hydrophilic oxygenated functional groups that allow the biomolecules to bind through both covalent and noncovalent interactions.

A DDS is able to deliver therapeutic drugs inside the affected cells in a controllable way, reducing the side effects on healthy tissues. GO, as a DDS, provides better biodistribution of drugs, reduced adverse effects on healthy cells, specific selectivity, and higher local therapeutic absorption. Due to these capacities, GO has been studied as a carrier for several drugs. For example, Wei et al. functionalized GO as a DDS for platinum anticancer drugs. The authors functionalized GO with high-molecular-weight polymers (polyethyleneimine, polyethylene glycol and chitosan) and folic acid for the accommodation of platinum anticancer drugs (Cisplatin, Carboplatin, Oxaliplatin and Eptaplatin). Because of the limitations of platinum anticancer drugs, such as the high protein binding rate, cross resistance, significant toxic side effects, and lack of specificity, the authors wanted to develop a drug carrier with good biocompatibility and high efficiency for drug delivery to the target site, reducing the side effects of platinum drugs. Wei et al. also analyzed the toxicity of drug delivery systems with GO in cell culture through the MTT assay. The results showed a viability rate of more than 80%, proving the low cytotoxicity of GO nanocarriers [70].

Another application of GO DDS in cancer therapy was presented by Abdelhamid and Hussein, who used Graphene Oxide as a DDS of methotrexate. The authors synthesized GO using Hummer’s method, and then methotrexate was loaded onto the GO surface via noncovalent interactions, including electrostatic and π-π interactions. They tested the drug delivery efficiency in hepatocellular carcinoma cells (HepG2 cells), human embryonic kidney cells (HEK293A cells), and porcine skin fibroblasts by incubating the cells with MTX-GO, MTX, or GO at 37 °C for 24 h. They evaluated the cell viability through the MTT assay and concluded that GO does not promote any side effects on the three cell types; however, the GO-MTX system induced significant cytotoxicity in the tumor cell line (HepG2 cells) compared to the normal cells (HEK293A and PEF cells), proving the efficiency of GO as a carrier [71].

There are many ways to control the release of the drug present in DDS—via pH, enzymatic interactions, Near-infrared (NIR) light, and electrical fields. Graphene functionalization can also play a role in controlling drug release.

There is evidence that the effects of photosensitizers on tumor cells are improved when they are conjugated with GO (as a carrier). This improvement is due to the large surface area and the availability of reactive functional groups on GO, which confers great efficiency to the delivery of photoactive drugs into tumor cells. Tumor sites have a characteristic acidic pH, and this promotes the release of anticancer drugs, since almost all GO-based delivery systems have pH-dependent behavior. To test this, Huang et al. developed a hybrid system with folic acid (FA) and GO-COOH. GO-COOH was produced by the reaction between GO, NaOH, and ClCH_2_COONa and was further neutralized and dialyzed [63]. With the complex GO-FA, it is possible to efficiently target the PS molecule Ce6. The authors loaded Ce6 onto the GO–FA complex with an efficiency of 80%, and the system was tested on human stomach cancer MGC803 cells. The results showed good cell internalization efficiency and high cytotoxicity of the hybrid system for phototherapies [2,72].

The focus of this review was on graphene as a base material for biomolecule immobilization and its application in biosensors, a promising field that covers different areas. Significant efforts are being made to improve the sensitivity and efficiency of biosensors that are crucial for medical analyses, as well as those that can be used for the detection of pathogens in food and environmental contaminants. The potential of graphene-based biosensors to detect infectious diseases is enormous, and the translation of this concept into clinical use would be revolutionary. However, with limitations, such as the ability of graphene to adsorb nontarget molecules, may cause false-positive detections, especially if noncovalent functionalization methods are applied. Additionally, the incorporation of biomolecules onto the graphene layer may be affected by external environmental factors like the temperature, pH, salt concentration, as well as the intrinsic properties of graphene itself. Future efforts should be directed toward improving the fabrication of biosensors with more cost-effective and easier techniques to achieve the desired biodevice. Future perspectives on sensor development should be directed toward the development of more portable, reproducible, miniaturized, and high-throughput detection devices.

## Figures and Tables

**Figure 1 nanomaterials-12-01624-f001:**
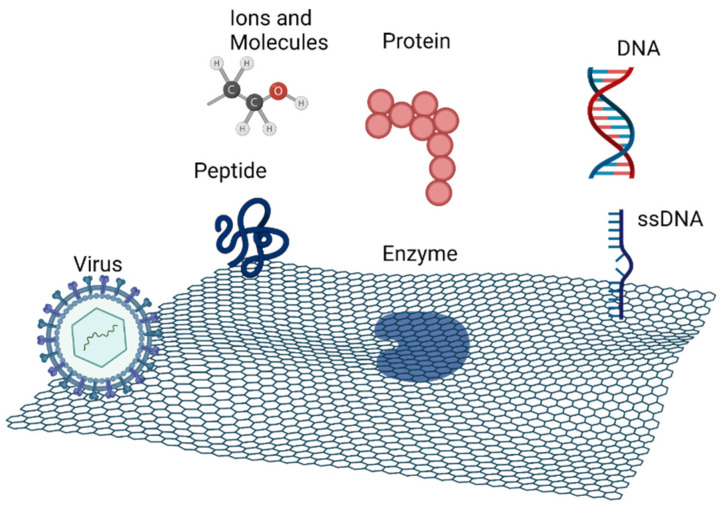
Schematic representation of the main biomolecular systems that can interact with graphene. Figure created with BioRender.com (accessed on 28 February 2022).

**Figure 2 nanomaterials-12-01624-f002:**
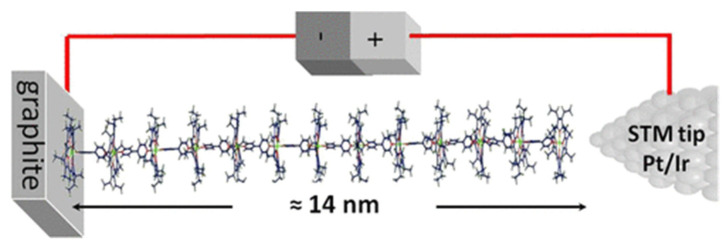
Self-assembled molecular porphyrin wire up to 14 nm in length. Republished with permission from The Journal of Physical Chemistry, from [30] Copyright 2022.

**Figure 3 nanomaterials-12-01624-f003:**
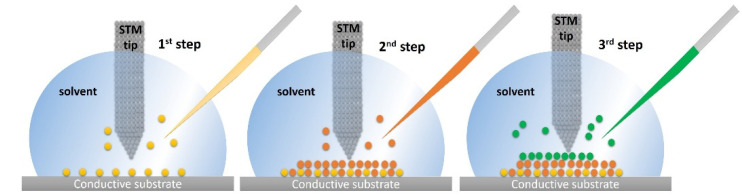
Illustration of a step-by-step method used to build supramolecular structures with STM at the solid/liquid interface. Reproduced from [31] with permission from the Creative Common CC BY license.

**Figure 4 nanomaterials-12-01624-f004:**
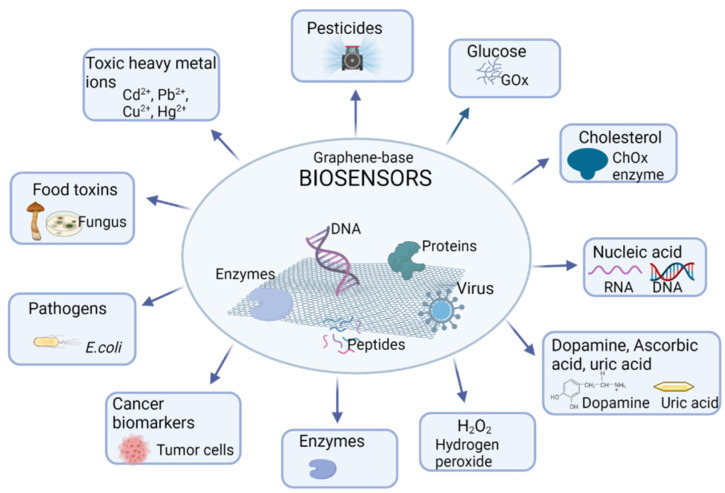
Applications of biosensors based on graphene and its derivatives. Created with BioRender.com (accessed on 2 March 2022).

**Figure 5 nanomaterials-12-01624-f005:**
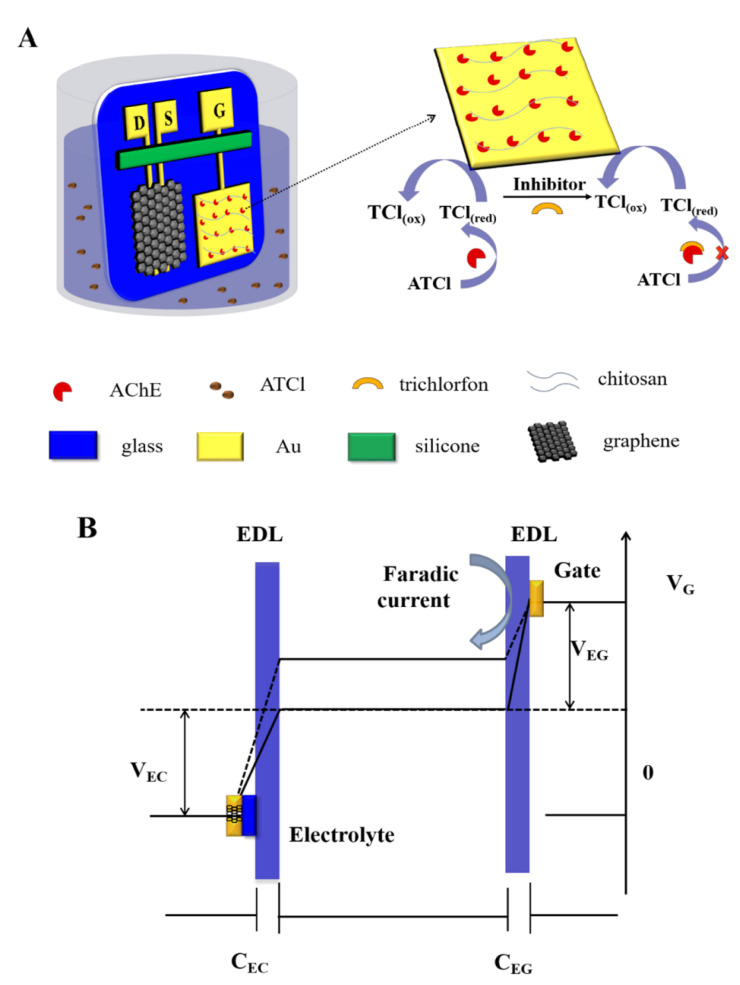
(**A**) Schematic representation of OP biosensor with immobilization of acetylcholinesterase (AChE) on a gold gate electrode with chitosan. D, S, and G represent the drain, source, and gate, respectively. (**B**) Representation of the potential drop between the gate and the graphene channel. Solid line represents the potential drop before the addition of trichlorfon in PBS solution, and dashed line represents the potential drop after the addition of trichlorfon in PBS solution. Reprinted with permission from [52]. Copyright© 2021 American Chemical Society.

**Figure 6 nanomaterials-12-01624-f006:**
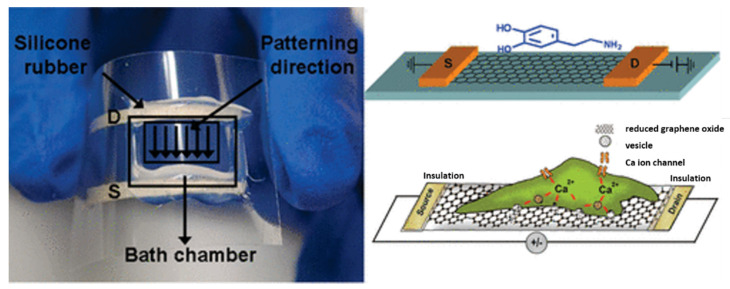
Representation of the rGO FET sensor for the detection of DA. Reprinted with permission from [54]. Copyright© 2010 American Chemical Society.

**Figure 7 nanomaterials-12-01624-f007:**
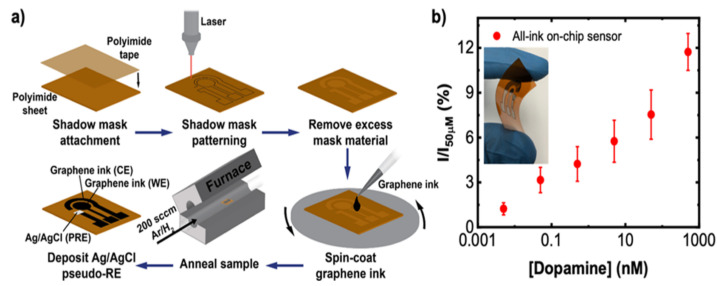
(**a**) Schematic representation of one-step thermal annealing process of graphene ink for a Dopamine sensor on a polyimide sheet. (**b**) Optical image of the sensor on a polyimide substrate. Graphic representation of the sensor response to DA in PBS solution. Reprinted with permission from [56]. Copyright© 2021 American Chemical Society.

**Figure 8 nanomaterials-12-01624-f008:**
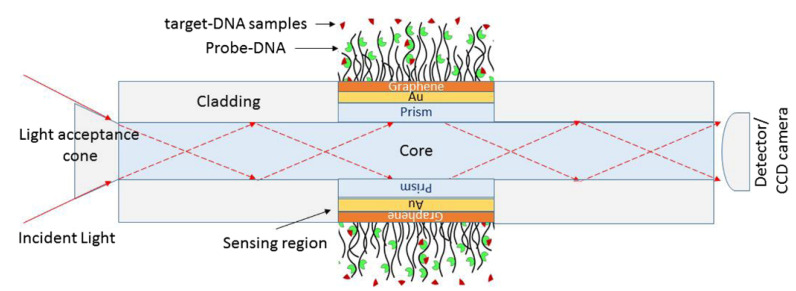
Illustration of an optic SPR biosensor: prism, Au, graphene, and phosphate buffer solution containing probe-DNA. Reprinted from [60] under the Creative Common CC BY licence.

**Figure 9 nanomaterials-12-01624-f009:**
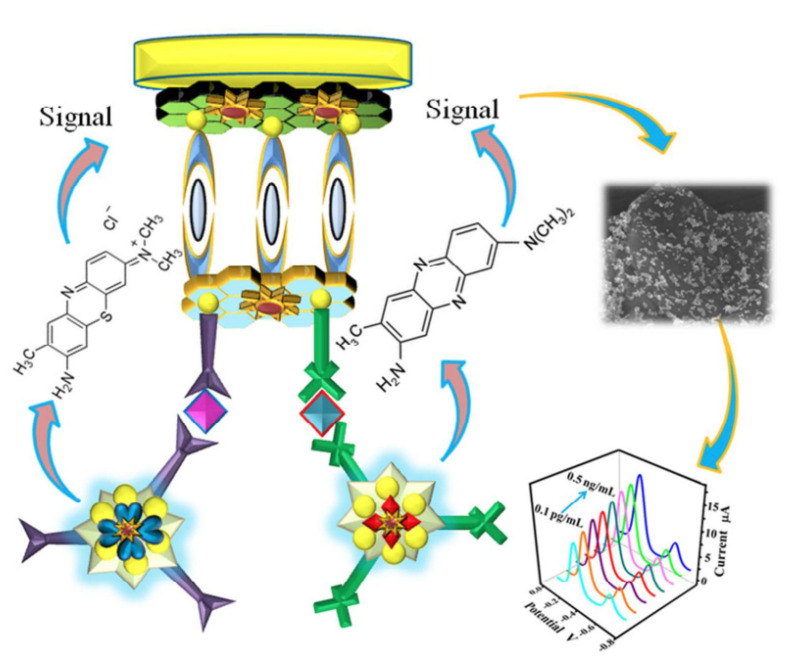
Representation of the tri-antibody dual-channel immunological biosensor for th edetection of cancer biomarkers. Reprinted with permission from [65]. Copyright©2017 American Chemical Society.

**Table 1 nanomaterials-12-01624-t001:** Interactions between biomolecules and graphene and the main applications of each biomolecule.

Biomolecule	Interaction		Application	Reference
	Covalent	Noncovalent		
DNA	Modified GO surface with organic-linker molecules	π-π, electrostatic, and hydrogen bonding	electrochemical DNA biosensors	Zhang and co-workers [7], Zhang et.al. [8], Wang et al. [9], Bo et al. [10]
Proteins	Functionalized hydrophylic GO surface achieved by hydrophobic aromatic groups on the surface	π-π, intermolecular forces, dipole–dipole interactions, and opposite charges	Biosensors	Hui et al. [11], Myung et al. [12]
Peptides	Functionalized GO surface	π-π stacking, electrostatic interactions	Biosensors with diagnostic functions	Kanchanapally et al. [13], Wu et al. [14], Wang and Lin [15], Li and co-workers [16]
Viruses	-	Hydrogen bonding, electrostatic interactions, redox reactions	Biosensors	Lee et al. [17], Oh et al. [18]
Enzymes	Free amino groups of enzymes bond with GO functionalized groups.	π-π stacking and electrostatic interactions	Biosensors	Unnikrishnan et al. [19], Li et al. [20], Liu et al. [21], Zhang et al. [22]
Porphyrins	−CO_2_H GO groups bond with –NH_2_ porphyrin groups	Electrostatic interactions	Cancer therapies	Santos et al. [23], Larowska et al. [24], Shim et al. [25], Ma et al. [26], Stefan-van Staden et al. [27]

**Table 2 nanomaterials-12-01624-t002:** Graphene-based biosensors with biomolecules as targets.

Target	Graphene-Base	Biosensor Type	Reference
Enzyme Bioreceptors			
Enzymes	GO-Nafion	Electrochemical	Cheragi et al. [41]
Glucose (GOx)	GCE/GO-Ph-AUNPS-CP/GOx	Electrochemical	Qi et al. [42]
Glucose (GOx)	SF/GOx-FET	FET	You et al. [43]
Glucose (GOx)	Graphene-Nafion/GOx	Electrochemical	Wei et al. [44]
Glucose (GOx)	GO-Gox	Electrochemical	Liu et al. [45]
Cholesterol (ChOx	Enzymatic–Ferroceno redox (Fc-GO)	Electrochemical	Dey et al. [46]
Cholesterol (ChOx)	Nonenzymatic–β-CD/GO	Electrochemical	Agnihotri et al. [47]
Glucose, Cholesterol, H_2_O_2_	TYR-GQDs	Fluorescent	Li et al. [48]
H_2_O_2_	MWCNT	Electrochemical	Settu et al. [49]
Pesticides (Ops)	PtNP-IML-PGE-GA	Electrochemical	Hondred et al. [50]
Pesticides (e.g., Flubendiamine, thiamethoxam, dimothoate, chlorpyrifos, dipterex)	GQDs-ABEI	Fluorescent	Gao et al. [51]
Pesticides (Ops)	Au-AChE-Graphene	Electrochemical	Wang et al. [52]
DA, AA, UA	functionalized multilayer graphene	Electrochemical	Srivastava et al. [53]
DA	rGO-FET-PC12cells	FET	He et al. [54]
DA	GO-based PCT	Fluorescent	Chen et al. [55]
DA	Graphene ink	Electrochemical	Butler et al. [56]
**Acid Nucleic Bioreceptors**			
Nucleic acid (DNA/RNA)	Graphene-Aunanorods-polythionine	Electrochemical	Huang et al. [57]
Nucleic acid (DNA)	AGNR-PBSAE	FET	Bagherzadeh et al. [58]
Nucleic acid	dsDNA-GO	Fluorescent	Xing et al. [59]
Anticancer drugs	Graphene	Optical	Wang et al. [60]
Pathogens (e.g., *E. coli* O157:H17)	GO-iron oxide-CS composite	Electrochemical	Tiwari et al. [61]
Pathogens (e.g., *E. coli* O157:H17	APTMS-ZnONR/c-GNF	Electrochemical	Jaiswal et al. [62]
Heavy metal ions (e.g., Cd^2+^, Pb^2+^, Hg^2+)^	L-cysteine-rGO	Electrochemical	Muralikrishna et al. [63]
Heavy metal ions (e.g., Ag^+^)	GO-Ag nucleotide	Fluorescent	Wen et al. [64]
**Antibody Bioreceptors**			
Cancer Biomarkers	Sulfur-doped GO-polyaniline	Electrochemical	Ren et al. [65]
Cancer Biomarkers (breast cancer cells)	Graphene-HER2	Electrochemical	Zahra et al. [66]
Food Toxins (e.g., Toxin AFB1)	rGO-antiAFB1-ITO	Electrochemical	Srivastava et al. [67]
Pathogens and food toxins (e.g., Toxin AFB1)	GOQDs	Fluorescent	Lu et al. [68]
Food Toxins (e.g., Histamine)	Graphene AJP	Electrochemical	Parate et al. [69]

## Data Availability

Not applicable.

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
