# Peer review of "Graphene Biosensors—A Molecular Approach"

_nanomaterials, 2022, doi:10.3390/nano12101624_

Round 1

Reviewer 1 Report

In this paper a molecular point of view on graphene sensors is provided in detail. The authors describe self-assembled biomolecular systems on top of graphite or graphene, self-assembled systems on graphene, and graphene bioapplications in biosensors ranging from electrochemical, over FET and plasmonic to optical biosensors. The paper is interesting and could serve as a nice reference for graphene sensors, however, some improvements need to be done before the manuscript can be recommended for publication.

Comments:

(1) conclusion is too short; at least give an overview of best performing devices, and critically examine areas where graphene is expected to have the greatest impact and have the greatest relevance

(2) section 6 adds very little in terms of future perspectives, so I suggest either to expand it significantly or to merge it with the conclusions section

(3) most references are 5+ years old; it would benefit the relevance of this review to find and include more recent literature on various graphene biosensors

(4) image quality for several figures needs to be improved

Author Response

Ms. Ivana Kalajdžić

Assistant Editor

The authors thank for the careful examination of the article made by the reviewer 1. We consider that the major revisions identified by the reviewer improve the paper and we addressed all of them. In the table of the uploaded file, we answer each point, and we include the main changes made to the manuscript. 

Reviewer 2 Report

The review entitled "Graphene Biosensors – A Molecular Approach" by 
Mónica Machado, Alexandra M. L. Oliveira, Gabriela A. Silva , Diogo B. Bitoque Joana Ferreira, Luis A. Pinto and Quirina Ferreira about the self-assembled monolayers of biomolecules on top of graphite with applications in biodevices. Authors pay a special attention to the porphyrins systems adsorbed on top of graphite that are able to anchorage other biomolecules.

They collect very good interesting data for the readers of the area and good figures are shown. Moreover, the references are quite well-structured.

As authors said: "this manuscript, significant developments have been made in graphene-based nanomaterials, exploring its excellent properties and applications in a large variety of fields from medicine to environment." They "focused on the application of graphene-based nanomaterials in biosensors, a promising field that covers different areas".

A work well done.

Author Response

Ms. Ivana Kalajdžić

Assistant Editor

The authors are thankful to reviewer 2 for the careful examination and are pleased to know that he appreciated the article. No revisions were requested, however the authors decided to improve the document introducing significant changes in Sections 3 and 4, following requests from other reviewers.

Reviewer 3 Report

The review manuscript submitted by Q. Ferreira and coworkers entitled “Graphene Biosensors – A molecular Approach” intend to describe the state of the art in the functionalization of graphene (and related surfaces) with self-assembled monolayers for application in biosensors. This review also gives a special attention at porphyrins systems on top of graphite.

Overall, I think this manuscript is not very well organized and is lacking coherence/readability:

  • Statements in the introduction are not supported by any reference from the literature.
  • There is a constant confusion in the considered surface and titles of subsections are often misleading (section 2 mention graphite and describe mainly graphene oxide; section 3 refers to graphene and mostly graphite and GO examples are presented etc.)
  • I would organize section 4 by sensor target instead of splitting it by the type of technology of the device. It would avoid the fragmentation of similar information (glucose biosensors in section 4.1.1, 4.2.1 and 4.5.1; H2O2 biosensors in 4.1.3 and 4.5.1 etc.)
  • The style is a bit descriptive and deeper analyses and comment on the reviewed work would sometimes be appreciable. English could be improved as well.

Moreover, most of the sections refers to recent review articles that already covers the subject (See ref 4, 6, 34, 48 in the present manuscript). I have a hard time to see the added value of the present manuscript.

To my opinion, this manuscript does not meet the requirements for publication in Nanomaterials.

I would encourage the author to build their article in a different way in order to make it more attractive. I find the part dedicated to porphyrins systems quite interesting even if important works of leading team in the field have not been included: Kenji Matsuda (Kyoto University) and Steven De Feyter (KU Leuven) for instance. Considering the authors nice contributions to the topic, I would recommend to focus the review on these aspects.

Author Response

Ms. Ivana Kalajdžić

Assistant Editor

The authors thank for the careful examination of the article made by the reviewer 3. We consider that the major revisions identified by the reviewer improve the paper and we addressed all of them. In the table of uploaded file, we answer each point, and we include the main changes made to the manuscript.

Round 2

Reviewer 1 Report

The authors have done a thorough revision and the manuscript is now in much better shape. However, some additional minor corrections are needed.

(1) Several references (e.g. 6, 34, 36, 37...) do not contain correct information; for example, in several of those publisher instead of journal name is given. Please check all references for accuracy.

(2) Text on the right hand side of Fig. 6 is not readable.

(3) The Conclusions section is split into too many paragraphs, which makes it difficult to follow. Please consider joining paragraphs for clarity and readability.

(4) There are minor problems with English but I think that can be corrected during editing/proofing.

Author Response

The authors thank for the careful examination of the article made by the reviewer 1. In the table of the uploaded file, we answer to each point, and we include the main changes made to the manuscript.

Reviewer 3 Report

I recognize that the authors have significantly improved their article but I still consider that overlap with existing reviews is large. However, if the editor consider the present manuscript of interst to their readership I think it has now a sufficient quality and readability to be published after minor revisions.

- Reference 6 is still described as an Accepted Manuscript even if it date back to 2016. It should be updated with final volume and page number.

- The name of the journal is not mentioned for reference 34.

- line 352: "Feyter et al." should be replace by "De Feyter et al."

- Authors might consider adding another reference of this team:
J. Am. Chem. Soc. 2014, 136, 50, 17418–17421 (https://pubs.acs.org/doi/10.1021/ja510930z)

Author Response

The authors thank for the careful examination of the article made by reviewer 3. In the table of the uploaded file, we answer each point, and we include the main changes made to the manuscript.
